# Evaluating Amazon effects and the limited impact of COVID-19 with purchases crowdsourced from US consumers

Alex Berke[1]*, Dana Calacci[2], Alex Pentland[1,3], Kent Larson[1]

1 MIT Media Lab, Massachusetts Institute of Technology, Massachusetts, United States of America,
2 Penn State University, University Park, Pennsylvania, United States of America, 3 HAI, Stanford University, Stanford, California, United States of America

* aberke@mit.edu

## Abstract

We leverage a recently published dataset of Amazon purchase histories, crowd-sourced from thousands of US consumers, to study changes in online purchasing behaviors over time, how changes vary by demographics, the impact of COVID-19, and relationships between online and offline retail. This work provides a case study in how consumer-level purchases data can reveal purchasing trends beyond those available from aggregate metrics. For example, in addition to analyzing spending behavior, we develop new metrics to quantify changes in consumers' online purchase frequency and the diversity of products purchased, to better reflect the growing ubiquity of online retail. Between 2018 and 2022 these consumer-level metrics grew on average by more than 85%, peaking in 2021. We find a steady upward trend in individuals' online purchasing prior to COVID-19, with a significant increase in the first year of COVID, but without a lasting effect. Purchasing behaviors in 2022 were no greater than the result of the pre-pandemic trend. We also find changes in purchasing significantly differ by demographics, with different responses to the pandemic. We further use the consumer-level data to show substitution effects between online and offline retail in sectors where Amazon heavily invested: books, shoes, and grocery. Prior to COVID we find year-to-year changes in the number of consumers making online purchases for books and shoes negatively correlated with changes in employment at local bookstores and shoe stores. During COVID we find online grocery purchasing negatively correlated with in-store grocery visits. This work demonstrates how crowdsourced, open purchases data can enable economic insights that may otherwise only be available to private firms.

**Data availability statement:** All data and code used in this analysis are available via the following open repository: https://github.com/aberke/amazon-study.

**Funding:** The author(s) received no specific funding for this work.

**Competing interests:** The authors have declared that no competing interests exist.

# 1. Introduction

The rise of e-commerce, led by Amazon, is transforming consumer behavior and retail markets. According to the US Census Bureau, e-commerce sales accounted for 9.4% of total US retail in 2018, growing to 14% by the end of 2022 [1]. Amazon commands an estimated 38% of this online market share, dwarfing its closest competition with less than 7% [2]. This dominance has led to the popular use of the term "Amazon effects" to describe changes ranging from new online consumer habits [3] and the resulting logistics issues for retailers [4] to the shifting relationships between online and offline retail [5]. The US Census Bureau provides quarterly estimates on e-commerce market share, but the data are highly aggregated, provided voluntarily by an undisclosed sample of firms [1]. Detailed data needed to understand the transformative "Amazon effects" remain largely within companies like Amazon, inaccessible to researchers and the public.

We address this public knowledge gap by analyzing a unique dataset of Amazon purchase histories crowdsourced from thousands of US consumers [6]. We leverage the disaggregated nature of this dataset to produce consumer-level metrics, providing new insights into how online retail is impacting consumers' purchasing behaviors and the "Amazon effects" impacting offline retail. This work presents a case study in how disaggregated purchases data can reveal trends beyond those available from aggregate metrics, which future work can build upon.

For example, while consumer behavior has traditionally been tracked by expenditure, this risks conflating changes in consumer behavior and prices. We introduce metrics that quantify changes in consumers' online purchase frequency and product diversity, which we argue better reflect how the increasing dominance and convenience of online retail are changing consumer behaviors. We use these metrics to analyze relationships between consumer demographics and e-commerce growth, and uncover the limited effects of COVID-19 on pre-pandemic growth trends.

## 1.1. Related work

Previous research has investigated how online retail has led to more dynamic pricing [7], finding prices are updated more frequently for goods available on Amazon [8], as well as more uniform pricing across store locations [8,9]. Research has also shown the effects Amazon can have on local retailers and employment. A study using data from 2010 to 2016 found the rollout of Amazon fulfillment centers reduced sales and employment at geographically proximate retail stores [10]. Earlier work using data from Amazon's online books marketplace has found substitution effects between used and new books [11], and local bookstore openings and online sales [12]. We add to this literature with analysis beyond just books, by analyzing substitution effects in other retail sectors where Amazon has substantially invested.

Much more e-commerce research emerged from the COVID-19 pandemic, with speculation that pandemic related shocks, such as store closures and stay-at-home orders, could accelerate the growth of online retail [13–15]. However, much of this

research was limited to the pandemic period without studying long-term effects. For example, the primary report released by the US Census Bureau on pandemic economic impacts showed substantial increases in e-commerce sales, but the analysis was limited to 2020 changes [16]. This work provides follow-up analysis to help evaluate the lasting impacts of COVID disruptions, leveraging longitudinal purchases data spanning the pre-pandemic (2018) to post-pandemic (2022) periods.

### 1.2. Study objectives and contributions

The following study is organized around two main objectives.

First, we develop new metrics to analyze changes in online purchasing behaviors, and the impact of COVID-19, as well as differences across demographic groups. (See methods in Section 2.2 and results in Section 3.1.) This work fills a gap left by previous analyses that evaluated economic disruptions during COVID, without evaluating beyond COVID. Contrary to speculation, we find COVID had a limited impact on trends in Amazon users' purchasing behaviors: COVID significantly increased online purchasing temporarily, but purchasing behaviors in 2022 were no greater than the result of the pre-pandemic trend. We further show differences in purchasing behaviors between demographic groups and how COVID impacted these groups differently.

Second, we use the Amazon purchases data to evaluate substitution effects between online and offline retail. (See methods in Section 2.3 and results in Section 3.2.) In particular, we analyze three retail sectors in which Amazon substantially invested: books [17], shoes [18], and grocery [19]. We demonstrate substitution effects between Amazon and brick-and-mortar retail for each of these sectors, contributing to a larger discussion of the Amazon effect on local stores.

Overall, our analyses help reveal nuance in the relationship between e-commerce and the pandemic, its diverse growth across consumer groups, and substitution effects impacting offline retail.

## 2. Materials and methods

### 2.1. Purchases and demographics data

Our analyses use an open dataset containing purchase histories and user demographics collected from N = 5027 US Amazon.com users [6]. The data were crowdsourced via an online survey and published with users' informed consent. Each user in the data contributed an export of their purchase histories from January 2018 to November 2022. The data collection process was previously described in [20], with the dataset detailed in [6]. Here we briefly describe aspects of the data pertinent to our analyses.

**2.1.1. Amazon purchases.** Each row in the purchases dataset represents a purchase by a particular user and includes an order date, product code (ASIN/ISBN), product title, per-unit price and quantity, state the item was shipped to, and a category assigned by Amazon. Purchases are linked to a single user and their demographics data via a response ID. To better illustrate the dataset, we provide an example set of rows in SI Table S1 in S1 File.

**2.1.2. User demographics.** Users who contributed their purchase histories also reported their demographics through a survey. The demographics include gender, age group, household income group, race and ethnicity, household size, and their US state of residence in 2021. Demographics used in the analyses are summarized in SI Tables S2-S5 in S1 File. Users who reported gender outside the male/female binary or did not disclose income are dropped from analyses that use the sex and income variables. The survey allowed users to identify as multiple races; our analysis that uses race includes users in all race groups they selected, allowing users to be in multiple groups (see Fig 4).

### 2.2. Analyzing changes in purchasing behavior

Fig 1 provides a visual overview of the data and metrics we use to analyze changes in consumer purchasing behaviors, showing the following metrics computed from our sample data: total expenditure, the number of distinct products

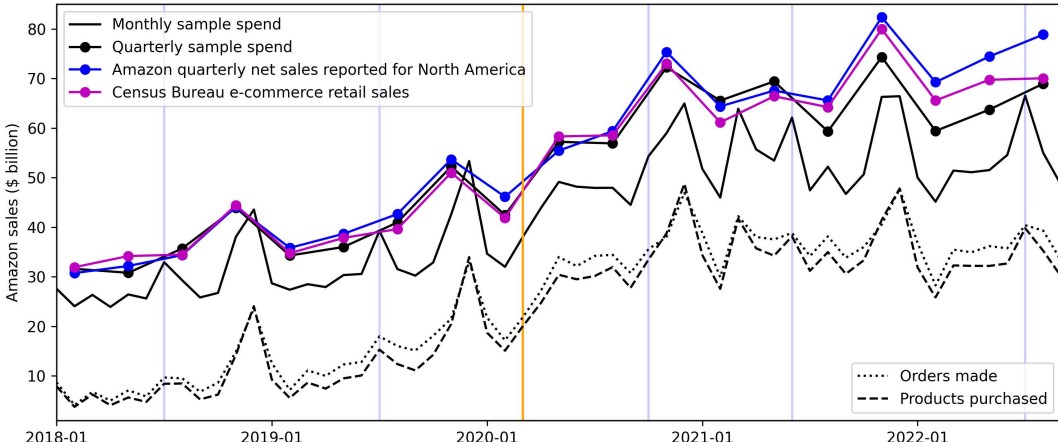

**Fig 1. Online purchasing metrics.** Quarterly Amazon net sales (North America segment) reported for investor relations and census e-commerce sales data, compared to metrics computed from our sample. Vertical blue lines indicate months Amazon Prime Day occurred. The orange line indicates March 2020, when COVID-19 had a major impact on US consumption. The sample metrics are scaled and shifted for legibility and should not be interpreted numerically.

purchased and the number of orders made (measured as per person purchase days). These metrics are shown alongside Amazon net sales data (North America segment), reported for investor relations [21], and e-commerce retail sales data from the U.S. Census Bureau [22], which are both reported on a quarterly basis. We use these quarterly sales data to validate our metrics and then use our metrics to help reveal details the quarterly sales data lack, including changes within quarters and how changes are driven by different consumer groups. When comparing our sample's quarterly expenditure to the Amazon net sales and census e-commerce sales data, there is a Pearson correlation of $r = 0.976$ ($p < 0.001$) and $r = 0.982$ ($p < 0.001$), respectively, indicating strong and consistent relationships between our sample data and the data used by the Census Bureau and reported by Amazon. Fig 1 also shows how our sample's expenditure grew less quickly than Amazon sales in later quarters, which is expected given our sample is limited to a consistent set of users while Amazon's user population grew over time. Vertical blue lines in Fig 1 indicate months when Amazon's major sales event, Prime Day, occurred [23]. We include these indicators throughout the results because Prime Day significantly increased the monthly metrics (see SI Table S6 in S1 File).

For the following analyses we use the metrics shown in Fig 1 computed as panel data.

**2.2.1. Preprocessing and panel data.** To produce the panel data used to estimate consumer behavior metrics, we first restrict the data to purchases made before November 2022. We filter out gift card purchases and restrict the data to users with purchases in both 2018 and 2022, allowing gaps in spending (e.g., months with zero spending). We then compute panel data with a row for each user-month, for each month in the span of 2018−01–2022−10. In order to estimate regressions that assume linear trends, we add a column variable, $t$, to the panel, numbering the months starting at $t = 0$ for 2018−01. Each row of panel data includes the monthly metrics for the corresponding user and month.

**2.2.2. Metrics.** We compute the following monthly metrics for each user-month in the panel data.

*Spend* is the total spend summed across users' purchases in USD.

*Distinct products* is the number of unique product IDs the user purchased.

*Purchase days* is the number of unique days the user made a purchase.

**2.2.3. Event study evaluating trends and the impact of COVID-19.** We produce a graphical event study in order to evaluate trends in consumers' purchasing behaviors and the impact of COVID. To do this, we first estimate

the following OLS regression model (Eq 1) using the panel data, which measures the average change in purchase days per month, per user, controlling for sex, age, and income demographics, where errors are clustered at the user level.

$$purchaseDays_{i,t} = intercept + \sum_t B_t \cdot T_{i,t} + \beta \cdot X_i + e_i \tag{1}$$

The term $X_i$ represents a vector of consumer variables including sex (male/female), age group, and income group; $T_{i,t}$ is an indicator variable for each time period, $t$. A coefficient, $B_t$, is estimated for each $t$. The estimated coefficients are in SI Table S11 in S1 File. We plot and further analyze the estimated $B_t$ coefficients (Fig 3).

We use the $B_t$ coefficients estimated for months 2018−01–2020−02 via Eq 1 in order to estimate a trend line prior to COVID. We do this via the following regression (Eq 2). Tabular results are in SI Table S12 in S1 File.

$$B_t = intercept + \beta \cdot t \tag{2}$$

We reproduce this analysis using *distinctProducts* and *spend* as the dependent variable in Eq 1, instead of *purchaseDays*.

We also test for whether there is a statistically significant increase in monthly purchasing via the following event study OLS model (Eq 3):

$$purchaseDays_{i,t} = intercept + \beta_1 \cdot t + \beta_2 \cdot postCOVID_t + \beta_3 \cdot X_i + \sum_m \beta_m \cdot month_t + e_i \tag{3}$$

We include an indicator variable for each month, $m$, to capture seasonality. $X_i$ represents a vector of consumer variables including sex (male/female), age group, and income group, and errors are clustered at the user level. The variable of interest is *postCOVID*, which is 1 if 2020−03 or later; 0 otherwise. We limit this event study to dates before 2021−03 in order to compare the period before COVID (2018−01–2020−02) to the first year impacted by COVID (2020−03–2021−02). As with the graphical event study, we repeat this analysis using *distinctProducts* and *spend* as the dependent variable. We also do an additional robustness check to help control for geographic variation, where we add US state of residence as an additional variable to Eq 3.

**2.2.4. Analyzing relationships between demographics and purchasing.** We evaluate relationships between consumer demographics and purchase frequency with the following OLS regression (Eq 4).

$$y_i = intercept + \beta \cdot X_i \tag{4}$$

The independent variables are consumer variables represented by vector $X_i$, specific to each user, $i$. The $X_i$ variables (shown in Fig 4 and Table S13 in S1 File) include sex, age group, income group, race and ethnicity, and household size. $X_i$ further controls for geography by including US state of residence. We run four separate OLS regressions, each using the same independent variables and only differing in the dependent variable, $y_i$. For (1) 2018 and (2) 2022 results, $y$ = purchase days per month, computed as the median over all months in the year, excluding November and December. Data are not available for November and December of 2022 (the data goes to November 2022), so we exclude November and December of 2018 as well in order to make the 2022 and 2018 data more comparable. We also evaluate changes over time: We (3) evaluate changes from 2018−2022, where $y$ = percent change in total purchase days in 2018 versus 2022, and we (4) evaluate changes over the first year of COVID, where $y$ = percent change in total purchase days from the year prior to COVID (2019−03–2020−02) to the period spanning the first year of COVID-19 (2020−03–2021−02).

## 2.3. Evaluating substitution effects between online and offline retail

In the following analyses we test for negative correlations which are consistent with substitution effects.

### 2.3.1. Online purchases versus retail employment prior to COVID.

To investigate locale-specific substitution effects prior to COVID, we test whether there is a negative correlation between changes in online purchases for category X and change in local employment at retail establishments of category X. We test this hypothesis for year-to-year changes in the years prior to COVID, 2018–2019, for X = books and shoes. To do this, we use yearly state level employment data for book stores and shoe stores retail establishments (NAICS codes 451211 and 448210, respectively) from the US Census Bureau Statistics of US Businesses (SUSB) tables [24].

We do this analysis for books and shoes because these products belong to well defined retail sectors where Amazon substantially invested and there are available employment data from the Census Bureau (i.e., there are corresponding NAICS codes). We limit the analysis to 2018 and 2019 because the purchases data starts in 2018, employment (SUSB) data were only available up to 2021 at the time of analysis, and the years 2020 and 2021 were impacted by COVID.

When computing changes in the Amazon purchases data by state, we use the shipping address state associated with each purchase. We restrict the analysis to users with purchases in 2018 and to states where at least n = 50 users made purchases. 45 states (including DC) meet this threshold. This excludes AK, MT, ND, PR, SD, VT, WY, and results in N = 4211 users across these 45 states.

We compute the correlation between (a) the percent change in the 2018–2019 employment for X = book/shoe stores in each state and (b) the percent change in the number of users making purchases for books/shoes in each state. For computing (a) the percent change in employment, we first normalize the number of employees for type X establishments in each state by the state population, specific to the year [25], in order to account for employment changes due to population changes. We then compute percent change as: (employees$_{2019}$ - employees$_{2018}$)/ employees$_{2018}$, where employees$_{2018}$ and employees$_{2019}$ are the normalized numbers. For (b) we first compute the number of distinct purchasers for category X in each state in each year as the mean of a repeated random sampling process. We randomly draw, without replacement, n = 2500 users, compute the number of distinct users buying X in each state for each year, and repeat the process 1000 times.

Finally, we then test for whether the Pearson correlation coefficient between (a) and (b) is negative and statistically significant at the $p < 0.05$ level, where a negative correlation is consistent with substitution effects.

### 2.3.2. Online grocery purchases versus in-store shopping during COVID.

A number of technology companies that collect location data from mobile phones produced mobility datasets for researchers to study the COVID-19 pandemic [26]. We use the Google COVID-19 Community Mobility Reports data [27] for "Grocery and pharmacy" produced for each US state, which measures how in-person visits and length of stay at these places changed compared to a baseline. (The baseline was the 5-week period of January 3 to February 6, 2020.) The data from Google is a daily metric, which we aggregate to a monthly metric by taking the monthly mean. We compare the monthly mobility metric to the monthly number of distinct users who made Amazon grocery purchases in the corresponding US state [28]. Grocery purchases were identified using the product categories listed in the SI.

We test for a negative correlation between the mobility metric, which represents presence at grocery stores, and the number of users purchasing groceries online, where a negative correlation is consistent with substitution effects. We test this relationship for all months where Google mobility data are available, 2020−02–2022−01, excluding the December months (2020−12, 2021−12) where the holidays increase shopping overall, both online and offline. We test the top 3 US states, ordered by the average number of monthly Amazon users purchasing groceries: CA, TX, NY. We note that when ordering by population, the top US states are (1) CA, (2) TX, (3), FL, (4) NY [28]. We use the number of purchasers rather than the US population (which excludes FL) partly because states like FL have a seasonal population, where a population influx can contribute to both in-store purchases (captured by the mobility index) and online purchases.

## 3. Results

### 3.1. Changes in consumer behavior

This section of the results presents analyses using the purchasing behavior metrics shown in Fig 1 computed as panel data: for each user, for each month, we computed total spend, the number of distinct products purchased, and the number of distinct days the user made purchases (purchase frequency). Fig 2 shows the distribution of user-level monthly metrics averaged over Q1 of each year (median), showing there is large variation in the metrics across individual users. Each metric grew on average by more than 85% between 2018 and 2022, peaking in 2021 (COVID-19 period). More details on these distributions are shown in SI Tables S6-S8 in S1 File and Fig S3 in S1 File.

**3.1.1. Relationships between spend, distinct products, and purchase days.** The purchasing behavior metrics are highly correlated, as shown in Fig 1. We briefly expand on this relationship and then focus the main analyses on monthly purchase days.

The Pearson correlation between monthly metrics for spend and distinct products is r = 0.757 (p < 0.001), for spend and purchase days is r = 0.702 (p < 0.001), and for purchase days and distinct products is r = 0.835 (p < 0.001). We also estimate a linear relationship between the monthly metrics for distinct products and purchase days. On average, 1 additional purchase day corresponds to approximately 2 additional products purchased each month ($\beta$ = 2.143; p < 0.001), with a small but positive interaction effect between time and purchase days ($\beta$ = 0.004; p < 0.001), indicating a positive trend over time. See SI Table S10 in S1 File for details.

**3.1.2. COVID-19 had a limited impact on the trajectory of online purchasing.** Fig 3 presents a graphical event study which we use to evaluate the impact of COVID on the upward trajectory of consumers' online purchase frequency. It displays the average change in purchase days per month, in a regression that controls for users' sex, age, and income demographics (Eq 1). The resulting coefficients are shown in Fig 3 relative to a 2018−01 baseline. Numerical results are in SI Table S11 in S1 File. We also estimated the positive trend prior to COVID (dashed line in Fig 3) via a linear regression (Eq 2) trained on the coefficients estimated for 2018−01–2020−02 (SI Table S12 in S1 File).

We do not find the COVID-19 pandemic had a lasting impact on the pre-pandemic trend. Instead, we find COVID provided a transient shock, significantly increasing online purchasing behavior above the trend line temporarily ($\beta$ = 0.5578; p < 0.001). Purchasing then returned to a level no higher than the pre-pandemic trend would have brought it to.

We use Eq 3 to complement the graphical event study, and test for the statistically significant increase due to COVID with numerical results, shown in Table 1. The estimated trend (t) indicates that, prior to COVID, when averaged across users, monthly purchase days increased on average by about 0.03 days each month ($\beta$ = 0.0326; p < 0.001), or by about 1

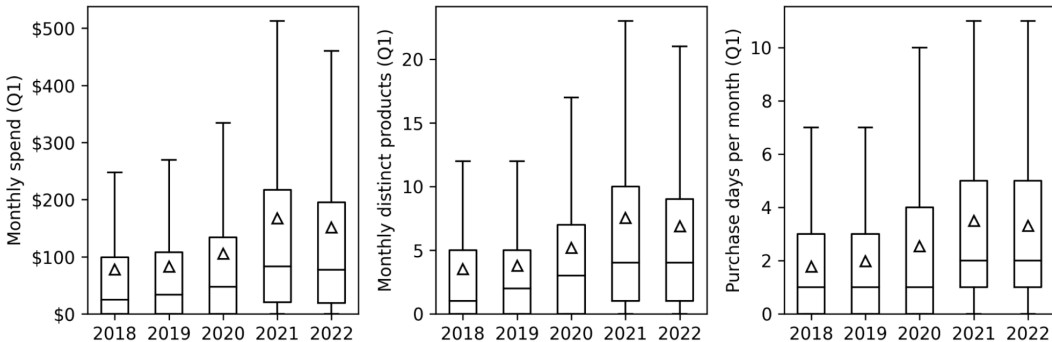

**Fig 2. Distribution of monthly metrics across users for Q1 of each year.** Boxplots show the medians (lines), means (triangles), first and third quartiles, and whiskers indicate the 1.5x IQR. Outliers are omitted (see SI Tables S6-S8 in S1 File).

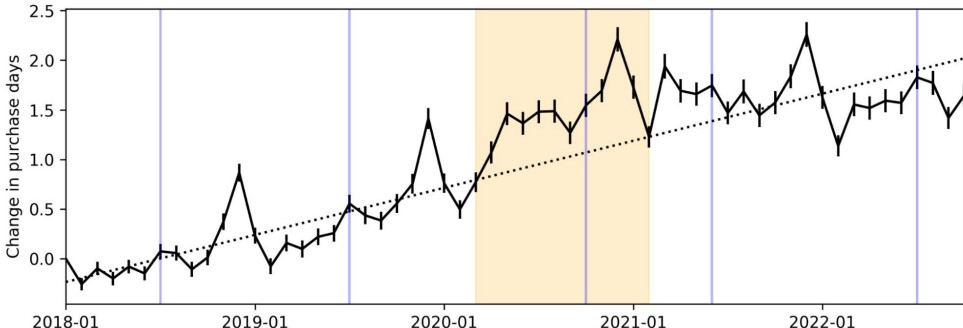

**Fig 3. Graphical event study estimating change in purchase frequency over time.** Solid lines display coefficients with 95% CIs. The dashed line displays the trend estimated over the pre-pandemic period (2018−01 to 2020−02). The orange section indicates the first year of COVID (2020−03 to 2021−02). Vertical blue lines indicate months Amazon Prime Day occurred.

day every 31 months. Whereas during COVID there was an average increase of more than 0.5 additional purchase days per month ($\beta = 0.5578$; p < 0.001) above this trend.

For robustness, we repeat the event study using the monthly number of distinct products purchased, and spend, as the dependent variable, instead of purchase days. The results are similar, showing a temporary boost due to COVID, where the metrics then resolve to the pre-pandemic trend line (see SI Fig S4-S5 in S1 File). When repeating the Eq 3 analysis using spend and distinct products as the dependent variables, we again find the post COVID coefficient is positive and statistically significant (p < 0.001). For robustness, we also repeat the Eq 3 analysis to include US state of residence, to help control for geographic variation, yielding the same coefficient and significance level for the post COVID variable as shown in Table 1.

**3.1.3. Purchasing behaviors differ by demographic groups.** Fig 4 shows results from the regressions analyzing relationships between demographics and purchasing (Eq 4), where statistically significant values (p < 0.05) are outlined in black. Numerical results are in SI Table S13 in S1 File.

We estimated four separate OLS regressions, each with a different dependent variable. The independent variables are the consumers' demographics. For results shown in the left panel of Fig 4, the dependent variables are consumers' median purchase frequency for 2018 and 2022. The right panel shows the relative differences between demographics when evaluating changes over time. The dependent variables are users' percent change in purchase frequency from 2018 to 2022, and percent change from the year prior to COVID to the first year of COVID.

Results show female consumers made online purchases more frequently than male consumers, and this gap grew from 2018 to 2022. However, we do not find the first year of COVID played a significant role in this growth.

Younger consumers (18–34 years) made online purchases significantly less often than middle aged (35–54 years) consumers and the oldest consumer group (55 years and older) increased their rate of purchasing less than their middle aged counterparts during the COVID period. And while consumers in the lowest household income group ($50k or less) purchased the least often, and the highest income group ($100k or more) purchased the most often, in both 2018 and 2022, purchase frequency for the highest income group grew the least from 2018 to 2022.

Race and ethnicity also played a role, with different purchasing behavior changes seen in the overall 2018–2022 study period versus the COVID period. Hispanic consumers substantially increased their purchase frequency from 2018 to 2022 compared to non-Hispanics, but this change was not significant during the COVID period. Black consumers did significantly increase their purchase frequency during COVID, compared to non-Black consumers, and this difference is large. We discuss potential reasons and implications for these differences in the Discussion.

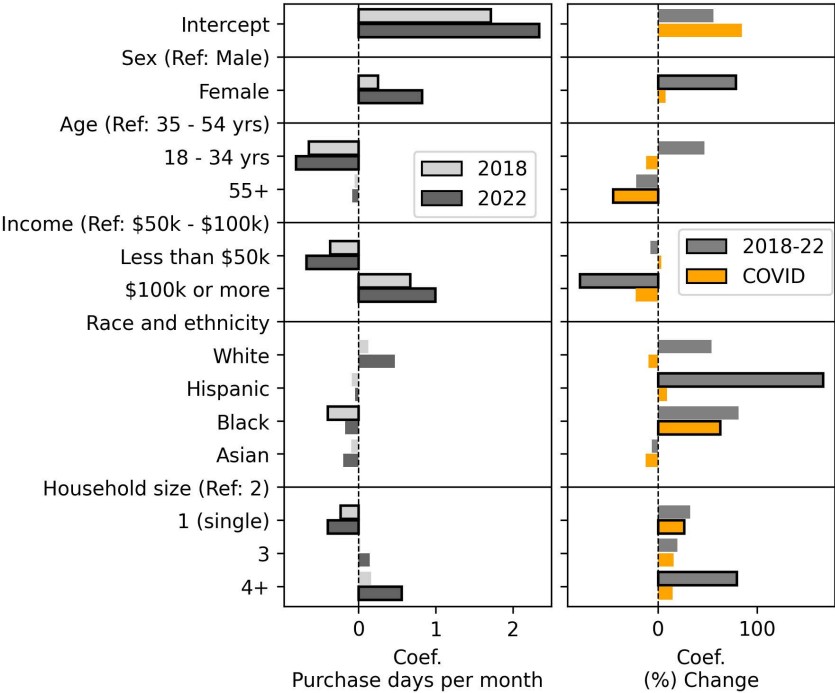

**Fig 4. Regression results showing relationships between purchasing and demographics.** Coefficients report estimated relative impact of consumer demographics on (left) purchase frequency for 2018 and 2022, and (right) change in purchase frequency from 2018 to 2022, and from one year prior to COVID to the first year of COVID. Bars indicating statistically significant values (p < 0.05) are outlined in black.

In addition we see an intuitive result where the smallest households purchase least often and the largest households purchase most often. From 2018 to 2022 the largest households (4 or more) saw the greatest increase in purchase frequency. In contrast, in the first year of COVID it was consumers living alone who increased their purchase frequency most.

### 3.2. Substitution effects between online and offline retail

While the previous results show changes in consumer purchasing behaviors, this section shows potential consequences for retail, by showing evidence of substitution effects between online and offline retail.

**3.2.1. Substitution effects for books and shoes prior to COVID.** We tested and validated a hypothesis that there is a negative relationship between online purchases and employment at local retail stores by leveraging yearly state level employment data. We compared year-to-year changes in the number of people making purchases for books and shoes in each state, to the change in employment at book stores and shoe stores in each state. We tested this hypothesis for years 2018–2019, prior to COVID, because we expect COVID disrupted how retailers made employment and planning decisions and employment data were not available in later years.

In this analysis, changes in employment broadly represents changes in availability of local (state) retail options as well as potential growth or contraction of employment within local retailers. We find the expected negative correlation with Pearson r = −0.344 (p = 0.021) and r = −0.315 (p = 0.035) for books and shoes, respectively, consistent with substitution effects.

This correlation does not imply a causal relationship in either direction. For example, it is possible more consumers make Amazon purchases for books and shoes when there are fewer local retail options available. It is also possible

**Table 1. Eq 3 event study regression results. The dependent variable is monthly purchase days.**

| Variable | Coef |
|---|---|
| Intercept | 2.0878 (0.089) |
| **Month (ref: 1)** | |
| 2 | −0.3667*** (0.019) |
| 3 | −0.3206*** (0.024) |
| 4 | −0.3071*** (0.024) |
| 5 | −0.1284*** (0.023) |
| 6 | −0.2043*** (0.023) |
| 7 | −0.0245 (0.024) |
| 8 | −0.1007*** (0.024) |
| 9 | −0.2780*** (0.022) |
| 10 | −0.1209*** (0.023) |
| 11 | 0.0819** (0.025) |
| 12 | 0.6069*** (0.026) |
| **Sex (ref: male)** | |
| Female | 0.4574*** (0.077) |
| **Age (ref: 35–54 years)** | |
| 18 - 34 years | −0.8400*** (0.084) |
| 55 years and older | −0.1442 (0.131) |
| **Income (ref $50,000 - $99,999)** | |
| $100,000 or more | 0.9046*** (0.107) |
| Less than $50,000 | −0.5181*** (0.082) |
| **t** | 0.0326*** (0.002) |
| **Post COVID** | 0.5578*** (0.035) |
| **Observations** | 150518 |
| **R-squared** | 0.089 |

Standard errors are reported in parentheses. ***, **, and * indicate statistical significance at the 1%, 5%, and 10% levels, respectively.

growth in Amazon purchases leads to reduced revenue and then reduced employment at local retailers. Either way, this negative relationship indicates substitution effects between Amazon and local book and shoe stores.

**3.2.2 Substitution effects demonstrated through COVID.** Again we tested and validated a hypothesis that there is a negative relationship between online and in-store shopping. Here we use data from the COVID-19 pandemic to test for such substitution effects with grocery purchases. The analysis takes advantage of how COVID-19 provided exogenous shocks, and how groceries necessitate frequent recurring purchases, versus other purchases that can more easily be delayed.

We used mobility data from Google reporting on visits to grocery stores in each state [27]. We compared the store visits data to the number of Amazon users making grocery purchases within each state. We tested the top 3 US states, ordered by the average number of monthly Amazon users purchasing groceries, CA, TX, NY. We find the expected negative correlation with Pearson $r = -0.544$ ($p = 0.007$), $r = -0.431$ ($p = 0.040$), and $r = -0.513$ ($p = 0.012$), for CA, TX, NY, respectively, where the negative correlations are consistent with substitution effects. We note these results are limited to the 3 states included in the analysis, and may not extend to the US more broadly. States in other regions may have different food environments or COVID policies, leading to different relationships in online versus offline purchasing.

## 4. Discussion

E-commerce and COVID-19 have had transformative effects on consumer behavior and retail markets, yet much of the data needed to study these changes remain within Amazon and other private firms. We begin to address this public knowledge gap by leveraging a unique, recently published dataset of Amazon purchase histories, providing new insights into how online purchasing has changed over time and the impact of COVID-19 on these trends. This work presents a case study in how consumer-level, crowdsourced purchases data have the potential to produce insights beyond those available from common data aggregates.

The itemized and longitudinal nature of the purchases data allowed us to develop new metrics, quantifying changes in consumers' purchase frequency and the diversity of products they purchase online. We argue that compared to expenditure, which prior analyses are limited to, these metrics better quantify changes in consumers' online purchasing behaviors. These consumer-level metrics increased persistently prior to COVID and throughout the pandemic, growing by more than 85% from 2018 to 2022 on average. We consider the rapid growth in these purchasing behaviors another kind of "Amazon effect". Our analyses also show nuanced differences in this growth across consumer demographics and between the overall study period versus the first year of the COVID pandemic. For example, female consumers in the dataset made online purchases significantly more often than their male counterparts, and increased their purchasing significantly more than males from 2018 to 2022, where the first year of COVID did not play a significant role in expanding this gap. The story is different when inspecting other demographic dimensions. The largest households purchased most often, and increased purchasing most from 2018 to 2022 compared to other household sizes, which may be intuitive. Yet single person households increased their purchasing most during the first year of COVID. Differences by race demonstrate how these differences may be important. Black consumers significantly increased their purchase frequency over non-Black consumers during COVID, and this difference is large. When considering why, we note that Black Americans suffered larger losses due to the pandemic [29] – they were overall hospitalized at 2x the rate of White Americans [30] – and early analyses by major news outlets brought attention to this racial disparity [31]. If this led to Black consumers adapting their shopping behaviors to reduce COVID risk factors, this may be reflected in their substantial increase in online purchase frequency during the pandemic. These results present an example of how disaggregated purchases data can help shed light on how economic and health related shocks may have differing impacts on various consumer groups.

Our analyses also fill in research gaps left from the pandemic period, where researchers and analysts projected COVID would accelerate e-commerce adoption, with the long-term impacts unknown [16–18]. We find COVID had a limited impact on the trajectory of the purchasing behaviors we studied. While our results show COVID significantly increased purchasing behaviors initially, we also find the metrics in 2022 were no greater than the result of the pre-pandemic trend.

The Amazon purchases also provide evidence of substitution effects between online and offline retail for the books, shoes, and grocery sectors. While our analyses do not present a causal relationship, previous researchers have, showing how the rollout of Amazon fulfillment centers reduced sales and employment for nearby retail establishments [11]. The researchers described these changes as e-commerce driven "creative destruction", as they also found corresponding employment gains in the transportation-warehousing and food services sectors, although these gains did not outweigh the losses for retail. Our analyses add to the larger discussion of Amazon's effect on local retail by showing how substitution effects occur in specific retail sectors where Amazon has made substantial investments. However, the limitations of our dataset limit our analyses to the state level and a small window of time. More local and causal analyses should be pursued to better understand these potential "Amazon effects".

### 4.1. Limitations and future work

These results are limited due to the limited timespan of the purchases data. It is possible purchasing behaviors plateaued, or it is possible that past 2022 they continued along the pre-pandemic trajectory. US Census Bureau data that reports

quarterly estimates of e-commerce retail sales [22] suggests e-commerce again trended upwards in the months following our study period (see SI Fig S1 in S1 File ). However, these census data track sales rather than purchasing behaviors, without controlling for prices and inflation, and they aggregate sales from a variety of undisclosed e-commerce firms, versus just Amazon. Further research would benefit from more crowdsourced Amazon purchases data.

In addition to timespan, our analyses are limited by the dataset's geographic granularity and user sample. For example, we use US states to evaluate and control for geographic differences, yet there is likely heterogeneity within states or across urban, suburban and rural areas. More work should be done to understand the role of geography in the transition from offline to online purchasing. With respect to the sample, the sampled users were already making purchases by 2018, which may leave gaps in the demographics analyses. Future work should explore how newer Amazon users engage with online versus offline retail. In particular, there will be a cohort of young consumers who came of age entering digital economies during COVID-19 lockdowns and their purchasing behaviors may be very different.

Furthermore, more robust and detailed analyses are limited by the dataset's sample size, which is small relative to Amazon's user base, which includes the majority of American consumers [32]. Even with these limitations, this work demonstrates how economic insights can be gained through open data crowdsourced from consumers, providing initial analyses for future researchers to build upon with more data.

## Supporting information

**S1 File.  Supporting Information [33–38].**
(PDF)

## Author contributions

**Conceptualization:** Alex Berke.

**Data curation:** Alex Berke.

**Formal analysis:** Alex Berke.

**Funding acquisition:** Alex Pentland, Kent Larson.

**Methodology:** Alex Berke, Dana Calacci.

**Software:** Alex Berke.

**Supervision:** Alex Pentland, Kent Larson.

**Visualization:** Alex Berke.

**Writing – original draft:** Alex Berke.

**Writing – review & editing:** Alex Berke, Dana Calacci, Alex Pentland, Kent Larson.

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
