## [Decision Letter · Decision Letter 0]

20 May 2025

Dear Dr. Berke,

Thank you for submitting your manuscript to PLOS ONE. After careful consideration, we feel that it has merit but does not fully meet PLOS ONE’s publication criteria as it currently stands. Therefore, we invite you to submit a revised version of the manuscript that addresses the points raised during the review process.

We look forward to receiving your revised manuscript.

Kind regards,

Zhifeng Gao

Academic Editor

PLOS ONE

Journal Requirements:

2. Please include captions for your Supporting Information files at the end of your manuscript, and update any in-text citations to match accordingly. Please see our Supporting Information guidelines for more information: http://journals.plos.org/plosone/s/supporting-information .

Reviewers' comments:

Reviewer's Responses to Questions

**Comments to the Author**

1. Is the manuscript technically sound, and do the data support the conclusions?

Reviewer #1: Yes

Reviewer #2: Yes

2. Has the statistical analysis been performed appropriately and rigorously?

Reviewer #1: No

Reviewer #2: Yes

3. Have the authors made all data underlying the findings in their manuscript fully available?

Reviewer #1: Yes

Reviewer #2: Yes

4. Is the manuscript presented in an intelligible fashion and written in standard English?

Reviewer #1: Yes

Reviewer #2: Yes

Reviewer #1: The paper examines the impact of online shopping platforms on consumers’ purchasing behavior and how this impact was influenced by the COVID-19 pandemic. It leverages a unique crowdsourced Amazon purchase dataset, which includes detailed information on consumers’ purchasing activities and demographic characteristics. Overall, I find the paper to be well written and the topic is interesting. However, I have several concerns regarding the empirical strategies employed in the study, which I outline below:

1. In Equation (3), you estimate the impact of the COVID-19 pandemic on the number of purchase days. I am curious about the rationale behind focusing solely on purchase days. Have you considered exploring other outcome variables such as the number of distinct products purchased or average spending? It would be helpful to know whether the pandemic had similar effects on these measures.

2. Also in Equation (3), have you considered controlling for geographic variation by including consumers’ state of residence in the regression? Online purchasing behavior could vary significantly across states due to differing policies, infrastructure, or local conditions during the pandemic.

3. In Equation (4), you analyze the relationship between consumer demographics and purchase frequency. The empirical strategy appears to rely on the assumption that “the 2018 and 2022 results are meant to be comparable” (line 205). Could you provide justification to support this assumption?

4. In the analysis of the relationship between online purchases and offline retail employment prior to the pandemic, you restricted the sample to states with at least 50 users making online purchases. What is the rationale for this cutoff? Could this restriction potentially bias the results by overestimating the impact of online purchases on offline retail employment?

5. In the same analysis, you estimate the relationship by calculating the correlation coefficient between online purchases and the change in employment within a sector, measured as (employees in 2019 – employees in 2018) / employees in 2018. However, employment changes could be driven by other factors, such as population shifts or dynamics in other sectors. Have you considered using a regression-based approach that controls for these potential confounding factors?

6. In analyzing the impact of online grocery purchases on in-store shopping during COVID-19, the analysis focuses only on the top three states with the highest average monthly number of Amazon users (CA, TX, and NY). I am concerned about the external validity of these results. States in the Midwest or other regions may have very different food environments and demographic characteristics, which could lead to different online and offline grocery shopping dynamics.

Reviewer #2: General Comments:

Thank you for the opportunity to review this paper titled “Evaluating Amazon effects and the limited impact of COVID-19 with purchases crowdsourced from US consumers”.

This paper presents a large amount of information on changes in consumer shopping behavior during the COVID-19 pandemic using crowdsourced Amazon data. While the dataset and research topic are unique and compelling, the manuscript currently lacks a coherent narrative, making it difficult to follow the presentation of findings.

The introduction should clearly set the direction of the study. However, as currently written, the discussion of research gaps and contributions is scattered across several paragraphs (e.g., the first, fourth, and fifth). A more structured approach is recommended—such as beginning with general background, clearly stating key research gaps, and explaining how this study addresses those gaps.

Currently, the literature review, gap identification, and contribution statements are mixed together throughout the section, making it hard for readers to form a clear understanding of the study's purpose. Additionally, the authors should include a statement of the overall study objective, followed by specific objectives.

In the Materials and Methods and Results sections, the authors should focus on a few core research questions rather than trying to cover all analyses. Key findings should be presented in the main manuscript—currently, some are placed in the Supplementary Information (SI), making it difficult for the reader to find the results. The use of numbered subsections could improve clarity, especially since multiple aspects of consumer behavior are being examined. Also, methodological reporting is inconsistent; some sections include model specifications and corresponding results, while others do not. These should be harmonized for clarity.

Finally, the Results section should discuss the findings in more detail, especially explaining the magnitude of effects, not only directional effects.

Specific comments:

Materials and Methods

User demographics: For Table S2, please clarify which Census data (year and dataset) was used.

Preprocessing and panel data: It’s unclear whether users were required to have made purchases in both 2018 and 2022. Does the panel require continuous activity, or are gaps (e.g., months with zero spending) allowed?

Event study evaluating trends in purchasing behavior and the impact of COVID-19

Equation 1 should be accompanied by a results table—not just a figure—to help interpret the results.

Table S11, the results include an intercept term, but it's unclear whether individual fixed effects were applied. Equation 3 has a_i (fixed effects for each user), but this isn't reflected in the results. Please clarify.

When interpreting postCOVID variable, in addition to directionality, please interpret the magnitude of the coefficient. For example, what does a coefficient of 0.5578 imply in practical terms?

Analyzing relationships between consumer demographics and purchasing

Why not use Equation 1 instead of a separate model for this analysis? Also, since multiple behavioral outcomes are analyzed for the same individual, Seemingly Unrelated Regressions (SUR) would be more appropriate to account for correlated errors.

Analyzing online purchases versus retail employment prior to COVID

The use of both (a), (b) and (1), (2) for itemizing points is confusing, especially since references are numbered as well. Consider using one consistent formatting style.

What is the meaning of correlation coefficients? Please explain their interpretation.

Results

If the magnitude of coefficients is being discussed in the text, the corresponding tables should be in the main manuscript rather than in the SI.

Indicate which model or equation each result refers to for clarity.

Provide more detailed interpretation of coefficients. Many results describe only the direction (positive or negative) without addressing effect size. The sentence “On average, 1 additional purchase day corresponds to approximately 2 additional products purchased each month” is a strong example of clear interpretation. Similar explanations should be included throughout the results section.

Overall, the results section needs better organization. Consider streamlining it according to the study’s main objectives to improve clarity and readability.

**Do you want your identity to be public for this peer review?** For information about this choice, including consent withdrawal, please see our Privacy Policy

Reviewer #1: No

Reviewer #2: No

---

## [Author Response · Author response to Decision Letter 1]

3 Jul 2025

Dear Reviewers,

We thank you for generously taking the time to review our manuscript and provide thoughtful feedback. Your comments have helped us clarify the objectives of our work and results, and improve the manuscript.

Please note we have performed new analyses, namely robustness checks, following your suggestions, and have updated our publicly available analysis code accordingly.

Please find our specific responses and changes for your comments below.

Reviewer #1

Comment 1

In Equation (3), you estimate the impact of the COVID-19 pandemic on the number of purchase days. I am curious about the rationale behind focusing solely on purchase days. Have you considered exploring other outcome variables such as the number of distinct products purchased or average spending? It would be helpful to know whether the pandemic had similar effects on these measures.

Response

Thank you for pointing out this discrepancy. Equations (1) and (2) produce a graphical event study, where we focus on purchase days, and then we repeat the analysis for distinct products and spend. Equation (3) tests for statistical significance corresponding to the graphical event study. Fig 3 and SI Fig S4-S5 show that the results are consistent across these dependent variables (purchase days, distinct products, spend). At your suggestion, we also repeat the Equation (3) analysis using distinct products and spend as dependent variables, and again find consistent results. We have added this to the manuscript.

Comment 2

Also in Equation (3), have you considered controlling for geographic variation by including consumers’ state of residence in the regression? Online purchasing behavior could vary significantly across states due to differing policies, infrastructure, or local conditions during the pandemic.

Response

Thank you for pointing out that geography can play an important role in consumers' online purchasing behaviors, where the resulting variation could impact the results in Eq 3.

Due to your comment, we did an additional robustness check, where we extended Eq 3 to include state of residence as a variable in the regression. We actually found the resulting coefficients and statistical significance for the postCOVID term and t (average monthly change) to be exactly the same with and without the state variable. We have added this robustness check to the manuscript. Thank you for this suggestion.

Please note that following our new analyses that handle your comments 2-3, we have updated our publicly available analysis code.

Comment 3

In Equation (4), you analyze the relationship between consumer demographics and purchase frequency. The empirical strategy appears to rely on the assumption that “the 2018 and 2022 results are meant to be comparable” (line 205). Could you provide justification to support this assumption?

Response

We believe there may be a misunderstanding due to how we wrote our methodology with unclear language. We have updated the language to make this clearer and provide justification (please see the revised section 2.2.4).

We now explain that: Data are not available for November and December of 2022 (the data only goes up to November 2022), so we exclude November and December of 2018 as well in order to make the 2022 and 2018 data more comparable.

Thank you for your helpful comment that led us to clarify our work.

Comment 4

In the analysis of the relationship between online purchases and offline retail employment prior to the pandemic, you restricted the sample to states with at least 50 users making online purchases. What is the rationale for this cutoff? Could this restriction potentially bias the results by overestimating the impact of online purchases on offline retail employment?

Response

We restricted the analysis to states with at least 50 users making online purchases to help reduce the potential impact of bias, or noise caused by states with few purchasers. E.g. since we make comparisons in percent changes between 2018 and 2019, a state with only 1 purchaser in 2018 that then gains more purchasers in 2019 could see a very large percent change. In other words, we filtered our data to states with more signal.

Comment 5

In the same analysis, you estimate the relationship by calculating the correlation coefficient between online purchases and the change in employment within a sector, measured as (employees in 2019 – employees in 2018) / employees in 2018. However, employment changes could be driven by other factors, such as population shifts or dynamics in other sectors. Have you considered using a regression-based approach that controls for these potential confounding factors?

Response

Thank you for helping us see that the way we wrote out our methods was unclear. We agree that population shifts must be accounted for. When computing change in employment within a sector, we first normalize the number of employees by the population for the corresponding state and year. We have updated the text to better explain this.

We also agree that other dynamics can contribute to changes in the retail markets we analyzed. However, our state-level data limits our ability to credibly account for these dynamics compared to studies that use fine-grained data to make causal claims about employment outcomes (Chava et al.). While we also evaluate employment outcomes, our goal is instead to identify broad trends, where our findings complement more precise, causal analyses by showing consistent patterns in sectors Amazon greatly invested in.

Comment 6

In analyzing the impact of online grocery purchases on in-store shopping during COVID-19, the analysis focuses only on the top three states with the highest average monthly number of Amazon users (CA, TX, and NY). I am concerned about the external validity of these results. States in the Midwest or other regions may have very different food environments and demographic characteristics, which could lead to different online and offline grocery shopping dynamics.

Response

Thank you for pointing out that we must address this limitation. We agree that states in the Midwest or other geographic regions may have encountered different online versus offline grocery shopping dynamics due to their different food environments. Unfortunately the data we use has too small a sample of users from most states to draw more conclusions. We have added an additional table to the SI (Table S5) to show the sample geographic distribution by US state.

To your point, our results can only speak to the states that were tested in our analysis. We have added an explanation of this limitation to the main text (Section 3.2.2), cautioning the reader that the results only apply to the three states we tested and may not extend to the US more broadly.

Reviewer #2

Comment 1

While the dataset and research topic are unique and compelling, the manuscript currently lacks a coherent narrative, making it difficult to follow the presentation of findings.

The introduction should clearly set the direction of the study. However, as currently written, the discussion of research gaps and contributions is scattered across several paragraphs (e.g., the first, fourth, and fifth). A more structured approach is recommended—such as beginning with general background, clearly stating key research gaps, and explaining how this study addresses those gaps.

Currently, the literature review, gap identification, and contribution statements are mixed together throughout the section, making it hard for readers to form a clear understanding of the study's purpose. Additionally, the authors should include a statement of the overall study objective, followed by specific objectives.

Response

Thank you for your insight. After reading your comments we realize you are quite right and have followed your suggestions by restructuring the introduction and adding a clear subsection for (1.1) Related work and (1.2) Study objectives and contributions.

Comment 2

In the Materials and Methods and Results sections, the authors should focus on a few core research questions rather than trying to cover all analyses. Key findings should be presented in the main manuscript—currently, some are placed in the Supplementary Information (SI), making it difficult for the reader to find the results. The use of numbered subsections could improve clarity, especially since multiple aspects of consumer behavior are being examined.

Response

Thank you for these comments. As suggested, we have numbered the sections and subsections to improve clarity. We have also reorganized the Materials and methods (Section 2) and Results (Section 3) to make clear we focus on two main objectives and corresponding research questions, which are introduced in the new subsection 1.2.

With respect to your other higher level comment about the presentation of results, we note you made more specific suggestions regarding this in the comments below. Please see below how we have made the corresponding changes.

Comment 3

Methodological reporting is inconsistent; some sections include model specifications and corresponding results, while others do not. These should be harmonized for clarity.

Response

Thank you for pointing out this inconsistency. We have made changes to improve consistency by only including model specifications in the methods sections and only including corresponding results in the results sections. To improve clarity, we have also made changes to indicate which model or equation results refer to, following your comment #14.

Comment 4

Finally, the Results section should discuss the findings in more detail, especially explaining the magnitude of effects, not only directional effects.

Response

Thank you for this high level note. Please see our changes described in response to your more detailed comments below.

Comment 5

User demographics: For Table S2, please clarify which Census data (year and dataset) was used.

Response

We updated the Table S2 description and caption with the Census data year and dataset citations.

Comment 6

Preprocessing and panel data: It’s unclear whether users were required to have made purchases in both 2018 and 2022. Does the panel require continuous activity, or are gaps (e.g., months with zero spending) allowed?

Response

Thank you for pointing out this should be clarified. We have added clarification that the filtering allows gaps in spending (e.g. months with zero spending).

Comment 7

Equation 1 should be accompanied by a results table—not just a figure—to help interpret the results.

Response

To briefly remind our reviewer who might be short on time: Eq 1 and Eq 2 are used to produce a graphical event study, shown in Fig 3. They are followed by the Eq 3 event study which complements the graphical event study by also estimating trend while testing for statistically significant changes in trend due to COVID. Eq 1 estimates more than 60 coefficients to help produce the graphical event study.

We have added the results table for Eq 1 to the SI (Table S10) and a reference to it from the main text. Since this table is excessively long, with more than 60 estimated coefficients, and the coefficients are represented in Fig 3, we assume you and readers would prefer it in the SI. In order to help better interpret these results without referring to the SI, we have also added the results table for Eq 3 to the main text, Section 3.1.2, as Table 1. We have added further description to Section 3.1.2 to help interpret these results, using numbers from Table 1.

Comment 8

Table S11, the results include an intercept term, but it's unclear whether individual fixed effects were applied. Equation 3 has a_i (fixed effects for each user), but this isn't reflected in the results. Please clarify.

Response

Thank you for identifying this discrepancy. We have updated Equations 1 and 3 and the accompanying text to clarify that our models use an intercept term and errors are clustered at the user level.

Comment 9

When interpreting postCOVID variable, in addition to directionality, please interpret the magnitude of the coefficient. For example, what does a coefficient of 0.5578 imply in practical terms?

Response

Thank you for this suggestion. We have added further description and interpretation to Section 3.1.2 (referenced in Comment 7). Namely, we describe how during COVID we find an average increase of more than 0.5 additional purchase days per month (β=0.5578; p<0.001) above the trend.

Comment 10

Analyzing relationships between consumer demographics and purchasing:

Why not use Equation 1 instead of a separate model for this analysis? Also, since multiple behavioral outcomes are analyzed for the same individual, Seemingly Unrelated Regressions (SUR) would be more appropriate to account for correlated errors.

Response

Equation 1 is designed to measure trend, while controlling for demographic variables. In contrast, Equation 4 allows us to make direct comparisons between demographic groups for specific time periods, and direct comparisons between time periods.

Comment 11

Analyzing online purchases versus retail employment prior to COVID: The use of both (a), (b) and (1), (2) for itemizing points is confusing, especially since references are numbered as well. Consider using one consistent formatting style.

Response

Thank you for pointing out this was confusing. We have removed the numeric itemization.

Comment 12

What is the meaning of correlation coefficients? Please explain their interpretation.

Response

Thank you for pointing out that further explanation would be helpful. For the correlation coefficients in Section 2.2, we have added an explanation that they indicate strong and consistent relationships between our sample data and the data used by the Census Bureau and reported by Amazon. For the correlation coefficients that we test for in Section 2.3, we added a note at the start of the section that we test for negative correlations which are consistent with substitution effects. We also added to the end of 2.3.1 (comparing changes in employment to changes in online purchases), explaining that we test for a negative correlation because a negative correlation is consistent with substitution effects. We added a similar explanation to 2.3.2 (comparing mobility data at grocery stores to online grocery purchases), and we added similar notes to the results presented in 3.2.1 and 3.2.2.

Comment 13

If the magnitude of coefficients is being discussed in the text, the corresponding tables should be in the main manuscript rather than in the SI.

Response

Please see our addition of Table 1.

Comment 14

Indicate which model or equation each result refers to for clarity.

Response

Thank you for pointing out this would add clarity. We now indicate when results correspond to Eq 1, Eq 2, Eq 3 and Eq 4.

Comment 15

Provide more detailed interpretation of coefficients. Many results describe only the direction (positive or negative) without addressing effect size. The sentence “On average, 1 additional purchase day corresponds to approximately 2 additional products purchased each month” is a strong example of clear interpretation. Similar explanations should be included throughout the results section.

Response

Thank you for this suggestion and example. Please see our additions to Section 3.1.2.

Comment 16

Overall, the results section needs better organization. Consider streamlining it according to the study’s main objectives to improve clarity and readability.

Response

Thank you for this suggestion. Based on your suggestions, we added subsection 1.2 to describe two main study objectives: first to analyze changes in consumer purchasing behavior and COVID-19, and second to analyze substitution effects between online and offline retail. We have reorganized both the methods and results sections around these two objectives. The first main objective is reported in results subsection 3.1 and the second main objective is reported in results subsection 3.2.

---

## [Decision Letter · Decision Letter 1]

15 Oct 2025

Dear Dr. Berke,

Thank you for submitting your manuscript to PLOS ONE. After careful consideration, we feel that it has merit but does not fully meet PLOS ONE’s publication criteria as it currently stands. Therefore, we invite you to submit a revised version of the manuscript that addresses the points raised during the review process.

We look forward to receiving your revised manuscript.

Kind regards,

Zhifeng Gao

Academic Editor

PLOS ONE

Journal Requirements:

Reviewers' comments:

Reviewer's Responses to Questions

**Comments to the Author**

Reviewer #1: All comments have been addressed

Reviewer #2: All comments have been addressed

2. Is the manuscript technically sound, and do the data support the conclusions?

Reviewer #1: Yes

Reviewer #2: Yes

3. Has the statistical analysis been performed appropriately and rigorously?

Reviewer #1: Yes

Reviewer #2: Yes

4. Have the authors made all data underlying the findings in their manuscript fully available?

Reviewer #1: Yes

Reviewer #2: Yes

5. Is the manuscript presented in an intelligible fashion and written in standard English?

Reviewer #1: Yes

Reviewer #2: Yes

Reviewer #1: Thank you for addressing my previous comments. I only have a few minor editing suggestions:

1. The abstract is somewhat lengthy. Please consider condensing it to improve readability.

2. In Table 1, please revise the footnote to read: “Standard errors are reported in parentheses. ***, **, and * indicate statistical significance at the 1%, 5%, and 10% levels, respectively.”

Reviewer #2: (No Response)

**Do you want your identity to be public for this peer review?** For information about this choice, including consent withdrawal, please see our Privacy Policy

Reviewer #1: No

Reviewer #2: No

---

## [Author Response · Author response to Decision Letter 2]

15 Oct 2025

Response to Reviewers

Evaluating Amazon effects and the limited impact of COVID-19 with purchases crowdsourced from US consumers

October 2025

Dear Reviewers,

Thank you again for generously taking the time to review our manuscript. Your thoughtful comments have improved this work.

Please see that we have addressed your final comments by reducing the number of words in the abstract and by updating the footnote in Table 1.

---

## [Editor Report · Decision Letter 2]

29 Oct 2025

Evaluating Amazon effects and the limited impact of COVID-19 with purchases crowdsourced from US consumers

PONE-D-25-03722R2

Dear Dr. Berke,

We’re pleased to inform you that your manuscript has been judged scientifically suitable for publication and will be formally accepted for publication once it meets all outstanding technical requirements.

**I noticed that some of the equations are not displayed appropriately in the PDF file, please make sure you correct the problem in your final submission. **

Kind regards,

Zhifeng Gao

Academic Editor

PLOS ONE
---

## [Editor Report · Acceptance letter]

PONE-D-25-03722R2

PLOS ONE

Dear Dr. Berke,

I'm pleased to inform you that your manuscript has been deemed suitable for publication in PLOS ONE. Congratulations! Your manuscript is now being handed over to our production team.

Kind regards,

on behalf of

Dr. Zhifeng Gao

Academic Editor

PLOS ONE